# Detection of *Mycobacterium tuberculosis* using Gene Xpert-MTB/RIF assay among tuberculosis suspected patients at Mizan-Tepi university teaching hospital, southwest Ethiopia: An institution based cross-sectional study

**Mengistu Abayneh**[1]*, **Murtii Teressa**[2]

1 College of Medical and Health Science, Department of Medical Laboratory Sciences, Mizan-Tepi University, Mizan, Southwest Ethiopia, 2 College of Medical and Health Science, Department of Medicine, Mizan-Tepi University, Mizan, Southwest Ethiopia

* menge.abay@gmail.com

**Data Availability Statement:** All relevant data are within the paper and its Supporting Information files.

## Abstract

### Background

Consistently deciding its current extent and chance elements of tuberculosis (TB) in all levels of clinical settings contributes to the anticipation and control exertion of the disease. In Ethiopia, updated information is still needed at every healthcare level and in different risk groups to monitor the national program's performance, which aims to attain the 2035 goal. Hence, this study aimed to generate additional evidence data on the magnitude of *Mycobacterium tuberculosis* using the Gene Xpert assay among TB-suspected patients at Mizan-Tepi university teaching hospital, southwest Ethiopia.

### Methods

A cross-sectional descriptive study was conducted from June to September 30, 2021. The required socio-demographic and other risk factor data were collected from a total of 422 suspected TB patients using a structured questionnaire. Approximately 392 pulmonary and 30 extra-pulmonary samples were collected and examined using the Gene Xpert-MTB/RIF assay. The statistical package for social sciences (SPSS) version 25 software was used to analyze the data.

### Results

In this study, *Mycobacterium tuberculosis* was detected in 12.5% (49/392) of pulmonary samples and 13.3% (4/30) of extra-pulmonary samples, giving an overall TB positivity of 12.6% (53/422). Rifampicin-resistant *M. tuberculosis* was detected in 3/53 (5.7%). Male sex (AOR: 2.54; 95% CI: 1.210, 5.354), previous contact (AOR: 4.25; 95% CI: 1.790, 10.092), smoking cigarette (AOR: 4.708; 95% CI: 1.004, 22.081), being HIV-positive (AOR: 4.27;

**Funding:** The authors received no specific funding for this work.

**Competing interests:** The authors have declared that no competing interests exist.

95% CI: 1.606, 11.344), and malnutrition (AOR: 3.55; 95% CI: 1.175, 10.747) were all significantly associated with *M. tuberculosis* detection using the GeneXpert MTB/RIF assay.

## Conclusion

The overall frequency of *M. tuberculosis* in this study was still significant in different risk groups, despite the proposed strategies, which aimed to reduce TB prevalence to as low as 10 per 100,000 populations by 2035. Early case detection with better diagnostic tools and public health measures are important prevention and control strategies to meet the proposed target and reduce the burden of TB in the country.

## Introduction

Tuberculosis, or TB, is an airborne infectious disease caused by *Mycobacterium tuberculosis*. The TB infection occurs when a person inhales droplet nuclei containing *Mycobacterium tuberculosi*s organisms and typically affects the lung, which is called pulmonary TB, but it can also affect other sites (extra-pulmonary TB). Since the infection dose of *M. tuberculosis* is very low, inhalation of fewer than 10 bacilli can lead to a new infection [1, 2].

Although TB is one of several preventable diseases, it is the 13[th] leading cause of death and the second leading infectious killer after COVID-19 (above HIV/AIDS). Currently, the COVID-19 pandemic threatens to reverse recent progress in reducing the burden of TB disease and improving access to care. The most obvious impact is a large global drop in the number of people newly diagnosed with TB and reported. This fell from 7.1 million in 2019 to 5.8 million in 2020. Reduced access to TB diagnosis and treatment has increased TB deaths. Best estimates for 2020 are 1.3 million TB deaths among HIV-negative people (up from 1.2 million in 2019) and an additional 214 000 among HIV-positive people (up from 209 000 in 2019), with the combined total back to the level of 2017 [3, 4].

TB is present in all countries and in all age groups. Especially in developing countries due to delayed diagnosis, weak laboratory services, and health care, high-risk co-morbidities, and the challenges of drug-resistant TB (DR-TB) cause millions of people to suffer and die from the disease each year. For instance, although, Africa accounts for only 16.72% of the world's population, an estimated 25% of existing TB cases and 34% of related deaths in 2019 worldwide occur in the regions [5], and access to essential TB diagnostics, care, and treatment services are very limited [6, 7].

Ethiopia is one of the 22 countries with the 8[th] highest TB burden globally and 3[rd] in Africa. According to the 2018 global TB report, there were 117,705 TB cases reported in the country and 28,600 died (3600 TB/HIV co-infected) [8]. The estimated incidence in 2019 was 140 cases per 100,000, resulting in an annual mortality rate of 23.5 per 100,000. The major causes of TB morbidity and mortality are a lack of access to early diagnosis and treatment services, co-infections with other infectious diseases such as HIV/AIDS [9, 10].

People infected with TB bacilli may not necessarily become sick with the disease, which can lay dormant for years. When someone's immune system is weakened, the chances of becoming sick are greater. Because people with active TB are a potential source of infection for others, updated data on prevalence and associated risk factors will provide valuable information to relevant agencies involved in TB prevention, management, and control. Hence, this study aimed to generate additional data on the magnitude of *Mycobacterium tuberculosis* using the Gene Xpert assay among TB- suspected patients at Mizan-Tepi university teaching hospital, southwest Ethiopia.

## Materials and methods

### Study area and period

This study was conducted at Mizan-Tepi University (MTU) teaching hospital from June to 30 September 2021. MTU teaching hospital is located in the southwest regional state and is found in Mizan- Aman town, at a distance of 561 km from Addis Ababa, the capital city of Ethiopia. It is the second more than 139 bedded teaching hospital in the southwestern part of the country, serving approximately 5 million people from four zones such as BENCH- SHEKO, KEFA, SHEKA, and MAJANG as a referral center. The region has an estimated density of 1310 people per square kilometer or 13 persons per hectare. The annual patient flow of the MTU teaching hospital was about 72500. Only one TB clinic was available at the hospital, and annually, an average of 610 TB confirmed cases visited a TB clinic.

### Study design and study subjects

The institution based cross-sectional study design was conducted to determine the TB positivity rate using the GeneXpert assay in TB-suspected patients. Pulmonary and extra-pulmonary samples were collected from all TB presumptive patients aged ≥5 years. Those patients with clinical signs and symptoms of TB such as night sweats, loss of appetite, loss of weight, fever, and cough that lasts at least 2 weeks were considered presumptive TB cases. Those patients who started anti-TB treatment and were sent for follow-up and those who couldn't provide appropriate or sufficient specimens were excluded.

### Sample size determination

The required sample size is calculated using Cochran's genera formula for a single population proportion by assuming that a confidence interval of 95%, margin error of 5%, and the prevalence of PTB (p) is (50%) which may be gives as the maximum sample size for this study. By considering a non- respondent rate of 10%, the final sample size was 422. Patients were recruited prospectively in a consecutive manner until the estimated sample size of 422 was obtained.

### Demographic and clinical data collection

First, the questionnaire was adopted and modified from different literature in the English language and was translated into local languages by interviewers to facilitate the understanding of the interviewee and limit the bias of data collection. Trained health professionals collected the data related to socio-demographic, clinical, and other environmental risk factors using interviewer- based questionnaire. The study participants were informed about the purpose of the study and its objective. For children less than18 years of age, data was collected with the help of their parents and guardians. During data collection, the occupational status of children depends on their families.

### Laboratory sample collection

Those patients identified with signs and symptoms of tuberculosis were ordered to bring a single appropriate sample for the diagnosis of TB. Accordingly, about 3–5ml of sputum (n = 392) was collected with an appropriate specimen container according to the standard procedures for the Gene Xpert technique [11, 12]. In the case of presumptive extra-pulmonary TB, lymph node aspirate (n = 12), pleural fluids (n = 7), gastric aspirate (n = 6), and cerebrospinal fluid (n = 5) were collected by physicians and considered in the analysis. The quality of the samples

was checked before the examination. For children, the samples were collected with the help of their parents and guardians.

## Sample processing

All the collected samples were processed in the laboratory as per standard procedures for the Gene Xpert technique (Cepheid) [11, 12]. In detail, pulmonary and extra-pulmonary samples were mixed with sample reagent (SR) and shacked manually 10 to 15 times, then incubated for 15 minutes at room temperature. Then 2 ml of the processed sample was transferred to the test cartridge, scanned the barcode, and loaded into the Gene Xpert instrument following the manufacturer's instructions. After 2 hours of sample loading, the results were interpreted by the Gene Xpert diagnosis system from the measured fluorescent signals and displayed automatically as detected, not detected, or invalid/error for MTB.

## Data analysis

Data generated from laboratory results and questionnaires of similar groups were collected together for analysis using the Statistical Package for Social Science (SPSS) window version 25.0. Frequency distributions and descriptive statistics such as the number and percent of patients were identified and calculated. A binary and multivariate logistic regression model was fitted to determine variables associated with TB positivity using the Gene Xpert technique. At a *p*-value of less than 0.05, variables were declared as to be significant predictors, and odds ratios with a 95% confidence interval were reported.

## Results

### Socio-demographic characteristics of study participants

A total of 422 TB-suspected patients were involved in the study. Of these, 60.0% (253/422) were male, and 45.5% (192/422) were age group of 16–34 years. More than half, 57.8% (244/422) of the participants were from rural residences, and 49.8% (210/422) with a formal education in primary school. Farmers accounted for 38.9% (164/422), and 58.3% (246/422) respondents were got an estimated monthly income of less than 50 US dollars (Table 1).

### Magnitude of tuberculosis

In this study, out of 422 participants who were suspected of having TB, 92.9% (392/422) were presumed to have pulmonary TB, and 7.1% (30/422) were presumed to have extra-pulmonary TB. *M.tuberculosis* was detected in 53 patients giving an overall TB positivity rate of 12.6%. Of the 53 TB- positive patients, 90.6% (48/53) were over 15 years of age. The detection of *M. tuberculosis* in patients presumptive for pulmonary was12.5% (49/392). *Mycobacterium* was detected in 13.3% (4/30) of extra-pulmonary specimens, of which 50.0% (2/4) were from lymph node aspirate. Among the total *M. tuberculosis* confirmed cases, rifampicin-resistant *M. tuberculosis* was detected in 5.7% (3/53) (Table 2).

### Factors associated with the detection of *M. tuberculosis*

The detection of *M. tuberculosis* was 14.6% (37/253) in males, which was 2.54 times (AOR: 2.54; 95% CI: 1.210, 5.354) more likely than in females, 9.5% (16/169). As compared to the reference age groups, *M. tuberculosis* was detected most frequently in the age group of 16 to 34 years, 13.5% (26/192), a difference, which was not statistically significant (*P*-value > 0.05) (Table 3).

**Table 1. Socio-demographic characteristics of study participants (n = 422).**

| Variables | Total N (%) | Results of Gene Xpert | |
|---|---|---|---|
| | | Detected (n = 53; 12.6%) N (%) | Not detected (n = 369; 87.4%) N (%) |
| **Gender** | | | |
| Male | 253 (60.0) | 37 (14.6) | 216(85.4)* |
| Female | 169 (40.0) | 16 (9.5) | 153 (90.5) |
| **Age in years** | | | |
| 5–15 | 39 (9.2) | 5 (12.8) | 34 (87.2) |
| 16–34 | 192 (45.5) | 26 (13.5) | 166 (86.5) |
| 35–60 | 118 (28.0) | 14 (11.9) | 104 (88.1) |
| > 60 | 73 (17.3) | 8 (11.0) | 65 (89.0) |
| **Residence** | | | |
| Urban | 178 (42.2) | 21 (11.8) | 157(88.2) |
| Rural | 244 (57.8) | 32 (13.1) | 212 (86.9) |
| **Marital status** | | | |
| Married | 260 (61.6) | 36 (13.8) | 224 (86.2) |
| Single | 162 (38.4) | 17 (10.5) | 145 (89.5) |
| **Variables** | **Total N (%)** | **Results of Gene Xpert** | |
| | | **Detected N (%)** | **Not detected N (%)** |
| **Education** | | | |
| No formal education | 120 (28.4) | 19 (15.8) | 101 (84.2) |
| Primary school | 210 (49.8) | 26 (12.4) | 184 (87.6) |
| Secondary and above | 92 (21.8) | 8 (8.7) | 84 (91.3) |
| **Occupation** | | | |
| Farmer | 164 (38.9) | 18 (10.8) | 146 (89.2) |
| Employee | 53 (12.6) | 5 (9.4) | 48 (90.6) |
| Merchant | 39 (9.2) | 6 (15.4) | 33 (84.6) |
| Students | 31 (7.3) | 4 (12.9) | 27 (87.1) |
| Daily labors | 71 (16.8) | 9 (12.7) | 62 (87.3) |
| Prisoners | 64 (15.2) | 11 (17.2) | 53 (82.8) |
| **Monthly income (US Dollars)** | | | |
| <50 | 246 (58.3) | 31 (12.6) | 215 (87.4) |
| 50–100 | 134 (31.8) | 17 (12.7) | 117(87.3) |
| >100 | 42 (10.0) | 5 (11.9) | 37(88.1) |
| **Family size** | | | |
| ≤ 4 | 163 (38.6) | 19 (11.7) | 144 (88.3) |
| ≥ 5 | 259 (61.4) | 34 (13.1) | 225(86.9) |

NB: EB:*: indicates variables with *p*-value < 0.05 in univariate analysis

The detection of *M. tuberculosis* was 15.0% (16/107) in those having a frequent alcohol-drinking habits, and 13.3% (10/75)in those having a Khat chewing habit, a difference, which was not statistically significant (*P*-value > 0.05). From those with a history of contact with known TB cases, *M. tuberculosis* was detected in 28.3% (13/46), which was4.25 times (AOR: 4.254; 95% CI: 1.790, 10.092) more likely to be positive for *M. tuberculosis* than those without contact history, 10.6% (40/376). *M. tuberculosis* was detected in 18.9% (17/90) of cigarette smokers, which was 4.71 times (AOR: 4.708; 95% CI: 1.004, 22.081) more likely to be positive for *M. tuberculosis* than those of non-cigarette smokers. The detection of *M. tuberculosis* was

**Table 2. Proportions of TB positive samples among presumptive TB patients at MTU teaching hospital, southwest Ethiopia, 2021.**

| Sample types | Total tested samples | Gene-Xpert status |
|---|---|---|
| | | Detected |
| Sputum | 392 | 49 (12.5%) |
| Lymph node aspirate | 12 | 2 (16.7%) |
| Pleural fluids | 7 | 1(14.3%) |
| Gastric aspirate | 6 | 1(16.7%) |
| Cerebrospinal fluids | 5 | 0 |
| **Total** | **422** | **53 (12.6%)** |

**Table 3. Socio-demographic factors associated with the detection of *M. tuberculosis* among TB suspected patients at MTU teaching hospital, southwest Ethiopia, 2021.**

| Variables | Total N (%) | Results of Gene Xpert | | COR (95% CI) | AOR(95% CI) | P-value |
|---|---|---|---|---|---|---|
| | | Detected N (%) | Not detected N (%) | | | |
| **Gender** | | | | | | |
| Male | 253 (60.0) | 37 (14.6) | 216(85.4) | 1.64 (0.879, 3.051) | 2.54(1.210, 5.354)* | 0.014 |
| Female | 169 (40.0) | 16 (9.5) | 153 (90.5) | 1 | 1 | |
| **Age in years** | | | | | | |
| 5–15 | 39 (9.2) | 5 (12.8) | 34 (87.2) | 1 | 1 | – |
| 16–34 | 192 (45.5) | 26 (13.5) | 166 (86.5) | 0.68(0.217, 2.113) | – | 0.502 |
| 35–60 | 118 (28.0) | 14 (11.9) | 104 (88.1) | 0.82(0.353, 1.916) | – | 0.650 |
| > 60 | 73 (17.3) | 8 (11.0) | 65 (89.0) | 0.91(0.364, 2.299) | – | 0.849 |
| **Residence** | | | | | | |
| Urban | 178 (42.2) | 21 (11.8) | 157(88.2) | 1 | | |
| Rural | 244 (57.8) | 32 (13.1) | 212 (86.9) | 0.89 (0.492,1.595) | – | 0.687 |
| **Marital status** | | | | | | |
| Married | 260 (61.6) | 36 (13.8) | 224 (86.2) | 1 | | |
| Single | 162 (38.4) | 17 (10.5) | 145 (89.5) | 1.37(0.742,2.532) | – | 0.312 |
| **Education** | | | | | | |
| No formal education | 120 (28.4) | 19 (15.8) | 101 (84.2) | 1.98(0.823,4.840) | – | 0.127 |
| Primary school | 210 (49.8) | 26 (12.4) | 184 (87.6) | 1.48(0.645,3.414) | – | 0.353 |
| Secondary and above | 92 (21.8) | 8 (8.7) | 84 (91.3) | 1 | | |
| **Monthly income** | | | | | | |
| < 50 | 246 (58.3) | 31 (12.6) | 215 (87.4) | 1.10(0.390.2.921) | – | 0.900 |
| 50–100 | 134 (31.8) | 17 (12.7) | 117(87.3) | 1.07(0.371,3.114) | – | 0.894 |
| >100 | 42 (10.0) | 5 (11.9) | 37(88.1) | 1 | | |
| **Occupation** | | | | | | |
| Farmer | 164 (38.9) | 18 (10.8) | 146 (89.2) | 0.59 (0.263,1.340) | – | 0.209 |
| Employee | 53 (12.6) | 5 (9.4) | 48 (90.6) | 1 | | |
| Merchant | 39 (9.2) | 6 (15.4) | 33 (84.6) | 0.88 (0.296,2.594) | – | 0.231 |
| Students | 31 (7.3) | 4 (12.9) | 27 (87.1) | 0.89 (0.296–2.594) | – | 0.811 |
| Daily labors | 71 (16.8) | 9 (12.7) | 62 (87.3) | 0.71(0.208,2.453) | – | 0.593 |
| Prisoners | 64 (15.2) | 11 (17.2) | 53 (82.8) | 0.70(0.269,1.816) | – | 0.463 |
| **Family size** | | | | | – | |
| ≤ 4 | 163 (38.6) | 19 (11.7) | 144 (88.3) | 1 | | |
| ≥ 5 | 259 (61.4) | 34 (13.1) | 225(86.9) | 0.87(0.480,1.490) | – | 0.657 |

NB

*: indicates variables with *p*-value < 0.05 in multivariate analysis

**Table 4. Environmental and clinical factors associated with the detection of *M. tuberculosis* among TB suspected patients MTU teaching hospital, southwest Ethiopia.**

| Variables | Total N (%) | Results of Gene Xpert | | COR (95% CI) | AOR(95% CI) | P-value |
|---|---|---|---|---|---|---|
| | | Detected N (%) | Not detected N (%) | | | |
| **Contact history with known TB** | | | | | | |
| Yes | 46 (10.9) | 13 (28.3) | 33(71.2) | 3.309(1.610,6.803)* | 4.25(1.790,10.092)* | 0.001 |
| No | 376 (89.1) | 40 (10.6) | 336(89.4) | 1 | 1 | |
| **Drinking raw milk habit** | | | | | | |
| Yes | 122 (19) | 17 (13.9) | 105 (86.1) | 1.187(0.639,2.206) | – | 0.587 |
| No | 300 (81.0) | 36 (12.0) | 264 (88.0) | 1 | | |
| **Smoking cigarette** | | | | | | |
| Yes | 90 (21.3) | 17 (18.9) | 73 (81.1) | 1.915(1.019,3.599)* | 4.708(1.004,22.081)* | 0.049 |
| No | 332 (78.7) | 36 (10.8) | 296 (89.2) | 1 | 1 | |
| **Drinking alcohol** | | | | | | |
| Yes | 107 (25.4) | 16 (15.0) | 91(85.0) | 1.224(0.643,2.328) | – | 0.538 |
| No | 315 (74.6) | 37 (11.7) | 278 (88.3) | 1 | | |
| **Khat chewing** | | | | | | |
| Yes | 75 (17.8) | 10 (13.3) | 65 (86.7) | 1.088(0.520,2.276) | – | 0.824 |
| No | 347 (82.2) | 43 (12.4) | 304 (87.6) | 1 | | |
| **Vaccination for BCG** | | | | | | |
| Yes | 51 (12.1) | 5 (9.8) | 46 (90.2) | 1 | | |
| No | 371 (87.9) | 48 (12.9) | 323 (87.1) | 0.731(0.277,1.932) | – | 0.528 |
| **HIV status** | | | | | | |
| Yes | 43 (10.2) | 14 (32.6) | 29 (67.4) | 4.209(2.051,8.636)* | 4.268(1.606,11.344)* | 0.004 |
| No | 379 (89.8) | 39 (10.3) | 340 (89.7) | 1 | 1 | |
| **Diabetes mellitus** | | | | | | |
| Yes | 29 (6.9) | 7 (24.1) | 22 (75.9) | 2.400(0.972,5.930) | – | 0.058 |
| No | 393 (93.1) | 46 (9.2) | 347 (90.8) | 1 | | |
| **Obstructive respiratory problems** | | | | | | |
| Yes | 21(5.0) | 6 (28.6) | 15 (77.4)* | 3.013(1.115,8.144)* | – | 0.082 |
| No | 401(95.0) | 47 (11.7) | 354 (88.3) | 1 | 1 | |
| **Malnutrition** | | | | | | |
| Yes | 55 (13.0) | 17 (30.9) | 38 (69.1)* | 4.113(2.110,8.017)* | 3.55(1.175,10.747)* | 0.025 |
| No | 367 (87.0) | 36 (9.8) | 331 (90.2) | 1 | 1 | |

NB

*: indicates variables with *p*-value < 0.05 in multivariate analysis

32.6% (14/43) in HIV-positive individuals, the odds of being positive for *M. tuberculosis* was 4.27 times (AOR: 4.268; 95% CI: 1.606, 11.344) much more than those of HIV-negative individuals. Similarly, the positivity of *M. tuberculosis* was more frequent in those individuals with malnutrition (MUAC<21cm), 30.9% (17/55), and the odds of being positive for *M. tuberculosis* was 3.55 times (AOR: 3.554; 95% CI: 1.175, 10.747) much more than those without malnutrition (Table 4).

## Discussion

This study was designed to detect *M. tuberculosis* using the Gene Xpert assay among tuberculosis-suspected patients in the southwest district of Ethiopia. Accordingly, the overall frequency of TB was 53 (12.6%), which is in line with the previous study finding in Ethiopia, in

which the average prevalence of TB was (12.6%) [13–16]. The result of this study was higher when compared with a study conducted at Jimma University Medical Center (9.3%) [17], Ataye district hospital (9.0%) [18], and Motta general hospital (8.4%) [19], North Ethiopia. However, this finding is lower than a study finding in Adigrat General Hospital (24.3%) [20] and Debre Markos referral hospital (23.1%) [21], Northern Ethiopia, and Gedeo Zone (26.8%) [22] and Adare general hospital (30.5%) [23], Southern Ethiopia. When comparing to other African countries, this finding is lower than a study finding in Morocco (20.6%) [24], Nigeria (22.9%) [25], Pakistan (38.0%) [26], Nepal (23.6%) [27] and other report from a developing country (45.3%) [28]. The observed variations in such studies may be related to variations in methodological techniques, study participants, study period, sample size, method of sample collecting, geography, and TB control and prevention practices. For instance, in this study, only the spot sample collection technique was used.

In this study, the positivity of *M. tuberculosis* was 12.5%and 13.3%in sputum and other tested extra-pulmonary samples, respectively. Another studies in Ethiopia also reported that the average prevalence of *M. tuberculosis* was 15.4% and 21.2% in pulmonary and extra-pulmonary TB cases, respectively [10, 17, 21]. A study in Morocco also reported that the positivity rates were 23.8% for pulmonary samples and 18.4% for extra-pulmonary samples, respectively [24]. Therefore, even though pulmonary TB is more researched and prevalent, extra-pulmonary TB also needs attention in the prevention and control of tuberculosis.

In this study, the detection rate of *M. tuberculosis* was significantly higher in males than females, which is also supported by many study reports in developing countries [21, 22, 26, 29, 30]. Although, it is not statistically significant, relatively higher prevalence of *M. tuberculosis* among males was reported in Addis Ababa, Ethiopia [16]. The reason for this might be due to social and behavioral deference which leads to higher exposure of males to crowded and outer environment for work reasons and other activities, such as smoking and alcoholism.

The present study also revealed that TB-HIV co-infection was found to be high (32.6%) and more likely to be positive for TB compared to patients who were HIV-negatives. This finding is in line with previous studies conducted across the country [17, 20, 22, 31, 32]. Similarly, malnutrition was found to be contributing factor to the positivity of *M.tuberculosis*. This finding is also supported by another study report in developing countries [33]. HIV and other co-morbidity conditions could reduce the ability of immune cells to fight infections and facilitate the development of active TB [31, 33–35]. Therefore, the efforts should be focused not only on the detection and treatment of infected individuals but also on the other contributing risk factors.

The other factor associated with the detection of *M. tuberculosis* infection in this study was contact history with known TB cases and cigarette smoking habit. In supporting this finding, many studies evidenced that, close contacts with patients with infectious tuberculosis and cigarette smoking are at increased risk of having *M.tuberculosis* infection and disease [21, 32, 36–40]. Smoking also damages the lungs and reduces the body's immune system, making smokers more susceptible to *M. tuberculosis* infection. The progression of latent to active tuberculosis has been directly associated with impairment of the immune systems and defects in immune cell responses [41, 42]. Therefore, early diagnoses with better diagnostic tools and treatment of known TB cases as well as reducing other contributing risk factors are important in the preventions and controlling of the transmission of TB infections.

## Limitations of the study

This study was a hospital based study with a small sample size which may not be representative of a total population. Rifampicin-resistant *M. tuberculosis* was not separately compared with other studies in the discussion as it is insignificant.

## Conclusion

Despite the overall frequency of *M. tuberculosis* being 12.6% in this study; further efforts may be needed with the proposed strategy by the government of Ethiopia to End TB by 2035, which targeted to reduce TB prevalence to as low as 10 per 100,000 populations. In addition, in this study, some socio-demographic, environmental, and clinical risk factors, such as male sex, contact history, being smokers, HIV infections, and malnutrition were found to be contributing factors for *M. tuberculosis* positivity by Gene Xpert- MTB/RIF assay. Therefore, these findings have a contribution to further combating *M. tuberculosis* infections, particularly in similar environments to those of the study population and site.

## Supporting information

**S1 File.**
(XLSX)

## Acknowledgments

First, we would like to thanks Mizan-Tepi University, Institute of Research and Community Support Coordinating Office. Thanks to all data collectors for their willingness during data collection.

## Author Contributions

**Conceptualization:** Mengistu Abayneh, Murtii Teressa.

**Data curation:** Mengistu Abayneh, Murtii Teressa.

**Formal analysis:** Mengistu Abayneh.

**Investigation:** Mengistu Abayneh.

**Methodology:** Mengistu Abayneh.

**Project administration:** Mengistu Abayneh.

**Supervision:** Mengistu Abayneh, Murtii Teressa.

**Validation:** Mengistu Abayneh.

**Visualization:** Mengistu Abayneh.

**Writing – original draft:** Mengistu Abayneh.

**Writing – review & editing:** Mengistu Abayneh, Murtii Teressa.

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
