## [Decision Letter · Decision Letter 0]

20 Sep 2022

PONE-D-22-20595Mycobacterium tuberculosis infection using Gene Xpert assay among tuberculosis suspected patients at Mizan-Tepi university teaching hospital, southwest Ethiopia: A cross-sectional studyPLOS ONE

Dear Dr. Abayneh,

Thank you for submitting your manuscript to PLOS ONE. After careful consideration, we feel that it has merit but does not fully meet PLOS ONE’s publication criteria as it currently stands. Therefore, we invite you to submit a revised version of the manuscript that addresses the points raised during the review process.

We look forward to receiving your revised manuscript.

Kind regards,

Sebsibe Tadesse, PhD

Academic Editor

PLOS ONE

Journal Requirements:

5. Please amend the manuscript submission data (via Edit Submission) to include author Murtii Teressa. 

Reviewer's Responses to Questions

**Comments to the Author**

1. Is the manuscript technically sound, and do the data support the conclusions?

Reviewer #1: No

Reviewer #2: Partly

2. Has the statistical analysis been performed appropriately and rigorously? 

Reviewer #1: No

Reviewer #2: Yes

3. Have the authors made all data underlying the findings in their manuscript fully available?

Reviewer #1: Yes

Reviewer #2: Yes

4. Is the manuscript presented in an intelligible fashion and written in standard English?

Reviewer #1: No

Reviewer #2: No

5. Review Comments to the Author

Reviewer #1: This is a observational study looking in proportion of patients with documented TB among patients with presumptive TB, and risk factors associated with the detection of TB in this group.

Study design: The problem is that this is a very biased population and the results do therefore not reflect the TB prevalence, but rather the positivity rate of GeneXpert in patients in whom a sample could be retrieved. All EPTB studied had a sample, only patients who can produc sputum are considered. It is unclear how the patients were included, based on a sample that was produced, are based on symptoms. For the latter it is unclear which symptoms were considered.

Sample processing: as Xpert MTB/RIF is a well-known method there is no need to put the laboratory procedures in such detail. Why the RIF resistance rate was not reported? In the discussion you say that it is insignificant, still it is OK to report it in the result.

Results:

English: several sections need heavy language editing, but especially the results.

Avoid using local currency if you mean to publish in international journals or add the equivalent in Dollars or Euro.

Table 2: it is surprising that only 7% of patients were vaccinated with BCG. How was this defined?

Table 1 and 4 can be combined, as can table 2 and 5.

Why are students selected as the reference population in table 4;

The OR are not correct. I did a check: if you compare farmers with students the OR = 0.83. If you compare farmers with all others the OR = 0.78.

Assume that the OR were correct, then the reporting is not correct: farmers are less likely to have a positive TB result, if their aOR = 0.37.

Reviewer #2: General comment:

This study investigated the prevalence of tuberculosis by genexpert TB detection system. I agree with the authors that it is very important to determine the prevalence the diseases at every level and different risk groups to monitor the national program’s performance to attain the 2035 goal in Ethiopia. It is institutional based study with its draw backs that it cannot be extrapolated to the general population in the study area. I suggest authors to address the flowing points in order to improve the manuscript.

Major and Compulsory Revisions:

1. The authors need language support in the writing.

2. Title: The title looks that the whole work of the article is dealing with infection with M. tuberculosis but not clinical TB. The authors need to make clear distinction between infection with M. tuberculosis and development of TB disease.

3. Abstract: Conclusion: The authors concluded that their finding reveals increased prevalence of M. tuberculosis infection. However, there was no supporting data trends that suggests the increased prevalence the disease overtime.

4. Introduction: The first sentence:” Tuberculosis, or TB, is an airborne infectious disease caused by Mycobacterium tuberculosis and is the leading cause of ill health and related deaths worldwide.” Is not true as it is against the first sentence of the second paragraph and currently available data.

5. Study area and period: Authors indicated that “Of the total annual patient visits, approximately 1080 people were visit a TB clinic.” But there was no indication about how many TB clinics are there at MTUH?

6. Laboratory sample collection

a. Authors described that “Those patients identified with signs and symptoms of tuberculosis were ordered to bring a single appropriate sample for the diagnosis of TB.” Which signs and symptoms of TB were considered?

b. There was no indication how the investigators identified cases that has started anti-TB treatment or not.

7. Sample processing: Which generations of cartridges were used?

8. Result: The authors described that “A majority of the respondents 164 (38.9%) were farmers”. How did the investigators determine majority?

9. Factors associated with Prevalence of TB: Authors reported that M. tuberculosis was detected in 18 (10.8%) of farmers, a proportion, which was 63.0% times greater than other occupations (AOR: 0.370; 95% CI: 0.143, 0.953). The data is wrongly described/presented as in the table the proportion of positive cases were more in groups for instance daily laborers and prisoners. The same wrong conclusion was taken in discussion as well.

10. Discussion

a. The authors indicate that “The result of this study was higher when compared with study conducted at Jimma University Medical Center (9.3%) [17], Southwest Ethiopia, Ataye District Hospital (9.0%) [18] and Motta general hospital (8.4%) [19], North Ethiopia”. However, their comparison did not take into account the 95% confidence around the figure (prevalence) reported by the authors.

b. They compared their findings with numerous published articles but did not discuss on individual basis why the difference was observed.

11. Minor Essential Revisions

a. Sample size: The use of p=0.5 is not justified as there are numerous studies from Ethiopia that can be used to calculate the more appropriate size.

b. How was malnutrition determined?

c. How was contact with known TB case was defined in terms of frequency and duration?

d. Result: Table 3 and figure 1 are repetitive

e. Ethical issues: the reference number of the ethical clearance by the competitive IRB should be indicated.

6. PLOS authors have the option to publish the peer review history of their article (what does this mean?). If published, this will include your full peer review and any attached files.

Reviewer #1: No

Reviewer #2: **Yes: **Gemeda Abebe

---

## [Author Response · Author response to Decision Letter 0]

26 Oct 2022

Thanks for your important and supporting comments

---

## [Editor Report · Decision Letter 1]

31 Oct 2022

Detection of Mycobacterium tuberculosis using Gene Xpert MTB/RIF assay among tuberculosis suspected patients at Mizan-Tepi university teaching hospital, southwest Ethiopia: An institution based cross-sectional study

PONE-D-22-20595R1

Dear Dr. Mengistu Abayneh,

We’re pleased to inform you that your manuscript has been judged scientifically suitable for publication and will be formally accepted for publication once it meets all outstanding technical requirements.

Kind regards,

Sebsibe Tadesse, PhD

Academic Editor

PLOS ONE

---

## [Editor Report · Acceptance letter]

10 Nov 2022

PONE-D-22-20595R1 

Detection of *Mycobacterium tuberculosis* using Gene Xpert-MTB/RIF assay among tuberculosis suspected patients at Mizan-Tepi university teaching hospital, southwest Ethiopia: An institution based cross-sectional study 

Dear Dr. Abayneh:

I'm pleased to inform you that your manuscript has been deemed suitable for publication in PLOS ONE. Congratulations! Your manuscript is now with our production department. 

Kind regards, 

on behalf of

Dr. Sebsibe Tadesse 

Academic Editor

PLOS ONE